# Gene Expression Profiles of Methyltransferases and Demethylases Associated with Metastasis, Tumor Invasion, CpG73 Methylation, and HPV Status in Head and Neck Squamous Cell Carcinoma

Larisa Goričan [1,†] , Tomaž Büdefeld [1,†] , Helena Čelešnik [1,2] , Matija Švagan [3] , Boštjan Lanišnik [3] and Uroš Potočnik [1,2,4,*]

1  Centre for Human Genetics and Pharmacogenomics, Faculty of Medicine, University of Maribor, 2000 Maribor, Slovenia; larisa.gorican@um.si (L.G.); tomaz.buedefeld@um.si (T.B.)
2  Laboratory for Biochemistry, Molecular Biology and Genomics, Faculty of Chemistry and Chemical Engineering, University of Maribor, 2000 Maribor, Slovenia
3  Department of Otorhinolaryngology, Cervical and Maxillofacial Surgery, University Medical Centre Maribor, Ljubljanska ulica 5, 2000 Maribor, Slovenia
4  Department for Science and Research, University Medical Centre Maribor, 2000 Maribor, Slovenia
*  Correspondence: uros.potocnik@um.si; Tel.: +386-2-2345-854
†  These authors contributed equally to this work.

**Abstract:** Epigenetic studies on the role of DNA-modifying enzymes in HNSCC tumorigenesis have focused on a single enzyme or a group of enzymes. To acquire a more comprehensive insight into the expression profile of methyltransferases and demethylases, in the present study, we examined the mRNA expression of the DNA methyltransferases DNMT1, DNMT3A, and DNMT3B, the DNA demethylases TET1, TET2, TET3, and TDG, and the RNA methyltransferase TRDMT1 by RT-qPCR in paired tumor–normal tissue samples from HNSCC patients. We characterized their expression patterns in relation to regional lymph node metastasis, invasion, HPV16 infection, and CpG73 methylation. Here, we show that tumors with regional lymph node metastases (pN+) exhibited decreased expression of DNMT1, 3A and 3B, and TET1 and 3 compared to non-metastatic tumors (pN0), suggesting that metastasis requires a distinct expression profile of DNA methyltransferases/demethylases in solid tumors. Furthermore, we identified the effect of perivascular invasion and HPV16 on DNMT3B expression in HNSCC. Finally, the expression of TET2 and TDG was inversely correlated with the hypermethylation of CpG73, which has previously been associated with poorer survival in HNSCC. Our study further confirms the importance of DNA methyltransferases and demethylases as potential prognostic biomarkers as well as molecular therapeutic targets for HNSCC.

**Keywords:** HNSCC; methyltransferase; demethylase; metastasis; invasion; HPV16

## 1. Introduction

Head and neck squamous carcinoma (HNSCC) is a group of cancers derived from the mucosal epithelium in the oral cavity, pharynx, and larynx [1]. With over 600,000 new confirmed cases annually, HNSCC is the eighth most common cancer worldwide [2]. People exposed to higher concentrations of carcinogens through occupational exposure or due to alcohol and nicotine abuse and those infected with human papillomaviruses (HPVs) are at higher risk of developing HNSCC, with alcohol consumption, smoking, and infection with HPV16/18 being the most common risk factors [3–7].

A hallmark of tumorigenesis is the reprogramming of DNA methylation patterns, involving the global loss of methylation and regional hypermethylation of specific genes [8–11]. Several studies have shown that extensive DNA hypomethylation and

hypermethylation in gene regulatory regions are associated with various clinicopathological characteristics of HNSCC, including tumor invasiveness and metastasis as well as risk factors such as tobacco and alcohol use and HPV infection [12–15]. The importance of DNA epigenetic modifications in HNSCC has been corroborated by studies on the expression and activity of DNA-modifying enzymes in tumorigenic processes. As an example, mRNA expression of DNA methyltransferase 1 (DNMT1), 3A (DNMT3A), and 3B (DNMT3B) was increased in tumors compared to normal tissue, and aberrant expression of DNMT1 showed a significant correlation with poor clinical outcomes and relapse-free survival in HNSCC patients [16]. In addition to DNA methyltransferases, HNSCC tumors exhibited decreased mRNA expression of the DNA demethylases Tet methylcytosine dioxygenase 1 (TET1) and 3 (TET3). Moreover, decreased TET3 expression was caused by methylation of the regulatory regions of TET3 and was positively correlated with poorer survival of HNSCC patients [17]. In another study, the expression of Tet methylcytosine dioxygenase 2 (TET2) was reduced in HNSCC patients, and the level of 5-hydroxymethylcytosine (5 hmC) was associated with decreased overall survival [18].

Although the gene-regulatory networks underlying cell invasion have been characterized, recent studies have shown that some genes, such as TIMP3 and PAX1, may also be under epigenetic control involving DNA methylation [19,20]. On the contrary, tumor invasion was found to be regulated by the hypomethylation of nuclear proteins, such as SPANXA1/2 and CLDN4 [21,22]. In addition to altered methylation patterns, changes in the expression of the methyltransferases DNMT1 and DNMT3B have also been reported in advanced tumors, further suggesting a role of epigenetic regulation in HNSCC invasiveness [23,24].

HPV infection of cells is another aspect of the pathobiology of HNSCC that is under epigenetic control. HPV(+) HNSCC is associated with persistent expression of the oncogenes E6 and E7, which, by targeting host tumor suppressors p53 and Rb, initiate neoplastic transformation [25–29]. In addition, both HPV E6 and E7, by acting directly or indirectly through p53, modulate the expression of DNMT1, 3A, and 3B, which consequently leads to epigenetic reprogramming [30–34]. Indeed, numerous studies have shown distinct genome methylation patterns in HPV(+) HNSCC compared to HPV(−) HNSCC associated with alcohol and tobacco consumption [35–39]. Although HPV(+) HNSCC has a better prognosis than HPV(−) HNSCC [40], the incidence of HPV(+) cancers is worrying, as the number of patients has increased over the years [41,42]. Further research is therefore needed into the molecular mechanisms involved in HPV-driven HNSCC.

Epigenetic events, such as genome-wide changes in DNA methylation patterns, play a critical role in the development and progression of HNSCC. Genomic methylation patterns result from the activity of various methyltransferases and demethylases affecting epigenetic reprogramming on a genome-scale or at specific genomic sites. To date, studies on the effect of methylation patterns in HNSCC have focused on a single enzyme or a group of enzymes, while data on the expression patterns of both methyltransferases and demethylases are lacking. Similarly, little is known about the epigenetic regulation of gene expression at the level of RNA methylation involving RNA methyltransferases. To determine the expression profile of methyltransferases and demethylases in HNSCC tumorigenesis, we performed an expression analysis of the DNA methyltransferases DNMT1, DNMT3A, and DNMT3B, the DNA demethylases TET1, TET2, TET3, and TDG, and the RNA methyltransferase TRDMT1 from the perspective of lymph node metastasis, invasion, and HPV16 infection in HNSCC patients. Furthermore, we previously found that CpG73 hypermethylation regulating miR-2682 expression is associated with tumor aggressiveness in HNSCC [43]. To further our knowledge of the epigenetic events controlling CpG73 methylation, we examined a correlation between the expression profile of the studied methyltransferases and demethylases and the level of CpG73 methylation.

## 2. Materials and Methods

### 2.1. Patients and Sample Collection

This study included a total of 34 newly diagnosed HNSCC patients before chemoradiation from a prospective cohort recruited between June 2015 and October 2019 at the University Medical Centre (UMC) Maribor, Slovenia. A pair of tumors and normal tissues were collected from each patient prior to pathological examination. Immediately after collection, fresh tissue biopsies were placed in RNAlater Tissue Storage Reagent (Invitrogen, Vilnius, Lithuania), incubated at 4 °C and −20 °C for 24 h each, and stored at −80 °C until further processing. Tumor staging was determined according to the 8th Edition TNM Classification for Head and Neck Cancer (UICC) by a pathologist. The study was approved by the Medical Ethics Committee of UMC Maribor (reference number UKCMB-KME11-5/15) and conducted according to the Declaration of Helsinki. Written informed consent for participation was obtained from all patients. The patient characteristics are summarized in Table 1. In terms of tumor invasion, metastasis, CpG73methylation, and HPV16 infection, mRNA gene expression did not differ significantly between the sexes. Therefore, female patients were not excluded from the analysis, and both male and female patients were considered as one experimental group. One pair of tumor–normal tissues was excluded from the analysis due to undetectable expression of the housekeeping genes ACTB and GAPDH.

**Table 1.** Patient characteristics.

| Clinicopathological Features | Men (*n* = 30) Number (%) | Women (*n* = 4) Number (%) |
|---|---|---|
| Age at diagnosis | 44–79 yrs. (mean 61.5 yrs.) | 50–65 yrs. (mean 57.8 yrs.) |
| Site | | |
| Lip and oral cavity | 4 (13.3%) | 1 (25.0%) |
| Pharynx | 20 (66.7%) | 2 (50.0%) |
| Larynx | 6 (20.0%) | 1 (25.0%) |
| Invasion [1] | | |
| Perivascular | | |
| Yes | 9 (30.0%) | 0 (0.0%) |
| No | 20 (66.7%) | 4 (100.0%) |
| Missing | 1 (3.3%) | / |
| Lymphovascular | | |
| Yes | 7 (23.3%) | 1 (25.0%) |
| No | 18 (60.0%) | 1 (25.0%) |
| Missing | 5 (16.7%) | 2 (50.0%) |
| Perineural | | |
| Yes | 8 (26.7%) | 0 (0.0%) |
| No | 21 (70.0%) | 4 (100.0%) |
| Missing | 1 (3.3%) | 0 (0.0%) |
| p16 status | | |
| Yes | 7 (23.3%) | / |
| No | 12 (40.0%) | 2 (50.0%) |
| Missing | 11 (36.7%) | 2 (50.0%) |
| Nicotine | | |
| Yes | 26 (86.6%) | 4 (100.0%) |
| No | 2 (6.7%) | / |
| Missing | 2 (6.7%) | / |
| Alcohol | | |
| Yes | 26 (86.6%) | 4 (100%) |
| No | 2 (6.7%) | / |
| Missing | 2 (6.7%) | / |
| Stage | | |
| pT (1–2) | 8 (26.6%) | 2 (50%) |
| pT (3–4) | 20 (66.7%) | 2 (50%) |
| Missing | 2 (6.7%) | / |

**Table 1.** *Cont.*

| Clinicopathological Features | Men (*n* = 30) Number (%) | Women (*n* = 4) Number (%) |
|---|---|---|
| pN (0) | 6 (20%) | 1 (25.0%) |
| pN (+) | 22 (73.3%) | 3 (75.0%) |
| Missing | 2 (6.7%) | / |

[1] Some tumors may have more than one type of invasion.

### 2.2. RNA Extraction and Quantitative Reverse Transcription PCR (RT-qPCR)

RNA was extracted from the tissues using Tri Reagent® (Sigma Aldrich, St. Louis, MO, USA) according to the manufacturer's instructions. Briefly, following homogenization, RNA was extracted from the tissue using phenol-chloroform-isopropanol extraction, washed two times in 1 mL of 75% ethanol, and dissolved in 50 μL of nuclease-free water. RNA concentration was determined using a Synergy 2 microplate reader (Biotek, Winooski, VT, USA), and RNA integrity was assessed using an Agilent Bioanalyzer 2100 (Agilent Technologies, Waldbronn, Germany) or by gel electrophoresis.

cDNA was prepared from 500 ng RNA using a High-Capacity cDNA Reverse Transcription Kit (Applied Biosystems, Vilnius, Lithuania) under the following thermal cycling conditions on a T Professional Basic Gradient thermal cycler (Biometra, Göttingen, Germany): 25 °C (10 min), 37 °C (120 min), and 85 °C (5 min). qPCR was carried out in a 10 μL reaction containing 1× SYBR green reaction buffer (LightCycler®480 SYBR Green I Master, Roche, Mannheim, Germany), 0.3 μM forward and reverse primer (each), nuclease-free water, and 2 μL of 1/20 diluted cDNA on a QuantStudio 12 K Flex Real-Time PCR System (Applied Biosystems, Singapore, Singapore). The thermal cycling conditions were: 95 °C (10 min) and 40 cycles: 95 °C (10 s), 60 °C (10 s), and 72 °C (20 s). The primer sequences of the studied genes are listed in Table 2. RT-qPCR was carried out in triplicate. Ct values > 40 were considered negative. Gene expression was normalized to the internal controls ACTB and GAPDH and quantified using the 2−ΔΔCt method.

**Table 2.** Primer sequences.

| Gene | | Primer Sequence |
|---|---|---|
| *DNMT1* | F: | 5′-AGCCGAGCGAGCCAGAGATA-3′ |
| | R: | 5′-CGTGTCAGAGATGCCTGCTT-3′ |
| *DNMT3A* | F: | 5′-ATGGAATCGCTACAGGGCTC-3′ |
| | R: | 5′-CTTCTGTGTGACGCTGCG-3′ |
| *DNMT3B* | F: | 5′-ATGGCAAGTTCTCCGAGGTC-3′ |
| | R: | 5′-CGATAGGAGACGAGCTTATTGA-3′ |
| *TET1* | F: | 5′-AGCTGTCTTGATCGAGTTATACA-3′ |
| | R: | 5′-CCCTTCTTTACCGGTGTACACTA-3′ |
| *TET2* | F: | 5′-CTGGCAAACATTCAGCAGCA-3′ |
| | R: | 5′-TTGAATTCAGCAGCTCAGTCC-3′ |
| *TET3* | F: | 5′-GGAACTCATGGAGGAGCGGTAT-3′ |
| | R: | 5′-GATCACAGCGTTCTGGCAGT-3′ |
| *TRDMT1* | F: | 5′-TCTCCAACCTCTCTTGGCATTC-3′ |
| | R: | 5′-GGAACTCCATCAGTACCTGACCA-3′ |
| *TDG* | F: | 5′-TGGACGTTCAAGAGGTGCAA-3′ |
| | R: | 5′-CTTAACTCCACGCTCTCAATTAGC-3′ |
| *ERBB3* | F: | 5′-TGAGGCGATACTTGGAACGG-3′ |
| | R: | 5′-TGGCCAGCATATGATCTGTC-3′ |

**Table 2.** *Cont.*

| Gene | | Primer Sequence |
|:---:|:---:|:---:|
| *ACTB* | F: | 5′-CATCGAGCACGGCATCGTCA-3′ |
| | R: | 5′-TAGCACAGCCTGGATAGCAAC-3′ |
| *GAPDH* | F: | 5′-TGAGAACGGGAAGCTTGTCA-3′ |
| | R: | 5′-CCCTGCAAATGAGCCCCA-3′ |

*2.3. Data Collection and Analysis*

Methylation data for CpG73 were obtained from a previous study. The DNA was isolated, and the percentage of methylated CpG73 was determined from the bisulfate-converted DNA as described previously [43].

Protein expression in HNSCC was determined by in silico analysis of the mass spectrometry data from the Clinical Proteomic Tumor Analysis Consortium (CPTAC) and the International Cancer Proteogenome Consortium (ICPC) datasets using UALCAN [44–46].

*2.4. Statistical Analysis*

Statistical analysis was performed using the Prism GraphPad software package (Prism 7.0, GraphPad Software, San Diego, CA, USA). The normal distribution of datasets was examined by the D'Agostino–Person omnibus normality test and the Shapiro–Wilk test. The effect of metastasis, invasion, and HPV16 on gene expression in normal and tumor tissue was examined by two-factor repeated measures ANOVA with tissue type and metastasis, tissue type and invasion or tissue type and HPV16 as independent factors, respectively. Sidak's multiple comparisons post hoc test was used to determine statistical differences between groups. To examine the correlation between the percentage of CpG73 methylation and gene expression, the first outliers were removed based on the Nalimov test ($\alpha = 0.05$) and the curated data were analyzed by Spearman's rank-order correlation. Statistical differences were considered significant at $p < 0.05$.

**3. Results**

*3.1. Tumors with Regional Lymph Node Metastases (pN+ Tumors) of HNSCC Showed Diminished mRNA Expression of DNMT1, DNMT3A, DNMT3B, TET1, and TET2*

To obtain a more comprehensive insight into the epigenetic control of tumor aggressiveness, we performed an RT-qPCR analysis of the DNA methyltransferases DNMT1, 3A, and 3B, the DNA demethylases TET1, 2, 3, and TDG, and the RNA methyltransferase TRDMT1 in relation to tumor metastasis and invasion. Tumors with regional lymph node metastases (pN+) showed higher expression of DNMT1 (effect of pN: $F_{(1, 29)} = 4.428$, $p < 0.05$; effect of tissue: $F_{(1, 29)} = 9.193$, $p < 0.01$; interaction pN x tissue: $F_{(1, 29)} = 11.67$, $p < 0.01$; Sidak's multiple comparisons test: $p < 0.01$), 3A (effect of pN: $F_{(1, 29)} = 11.52$, $p < 0.01$; effect of tissue: $F_{(1, 29)} = 6.458$, $p < 0.05$; interaction pN x tissue: $F_{(1, 29)} = 5.767$, $p < 0.05$; Sidak's multiple comparisons test: $p < 0.05$) and 3B (effect of pN: $F_{(1, 29)} = 3.194$, $p = 0.084$; effect of tissue: $F_{(1, 29)} = 14.07$, $p < 0.001$; interaction pN x tissue: $F_{(1, 29)} = 3.324$, $p = 0.079$; Sidak's multiple comparisons test: $p < 0.01$) than the paired normal tissue (Figure 1A–C). On the contrary, the expression of DNMT1, 3A, and 3B did not differ between tumor and paired normal tissue in pN+ tumors and was diminished in comparison to non-metastatic (pN0) tumors (DNMT1 and 3A: Sidak's multiple comparisons test, $p < 0.001$; DNMT3B: Sidak's multiple comparisons test, $p < 0.05$) (Figure 1A–C). Similarly, pN+ tumors showed significantly lower expression of TET1 (effect of pN: $F_{(1, 29)} = 6.068$, $p < 0.05$; effect of tissue: $F_{(1, 29)} = 6.254$, $p < 0.05$; interaction pN x tissue: $F_{(1, 29)} = 4.119$, $p = 0.052$; Sidak's multiple comparisons test: $p < 0.01$) and 3 (effect of pN: $F_{(1, 29)} = 3.658$, $p = 0.066$; effect of tissue: $F_{(1, 29)} = 3.578$, $p < 0.069$; interaction pN x tissue: $F_{(1, 29)} = 8.624$, $p < 0.01$; Sidak's multiple comparisons test: $p < 0.01$) in comparison to pN0 tumors and the expression of TET1 and 3 was significantly (TET1 and 3: Sidak's multiple comparisons test: $p < 0.05$) higher in pN0 tumors than paired normal

tissue (Figure 1D,F). The expression of TET1 and 3 did differ between tumor and paired normal tissue in pN+ tumors (Figure 1D,F). These data indicate that a distinct expression profile of DNA-methyltransferase/demethylases including a decreased expression of major the methyltransferases *DNMT1*, *3A*, and *3B* and the DNA demethylases *TET1* and *3* may be required for solid tumors to develop regional lymph metastases.

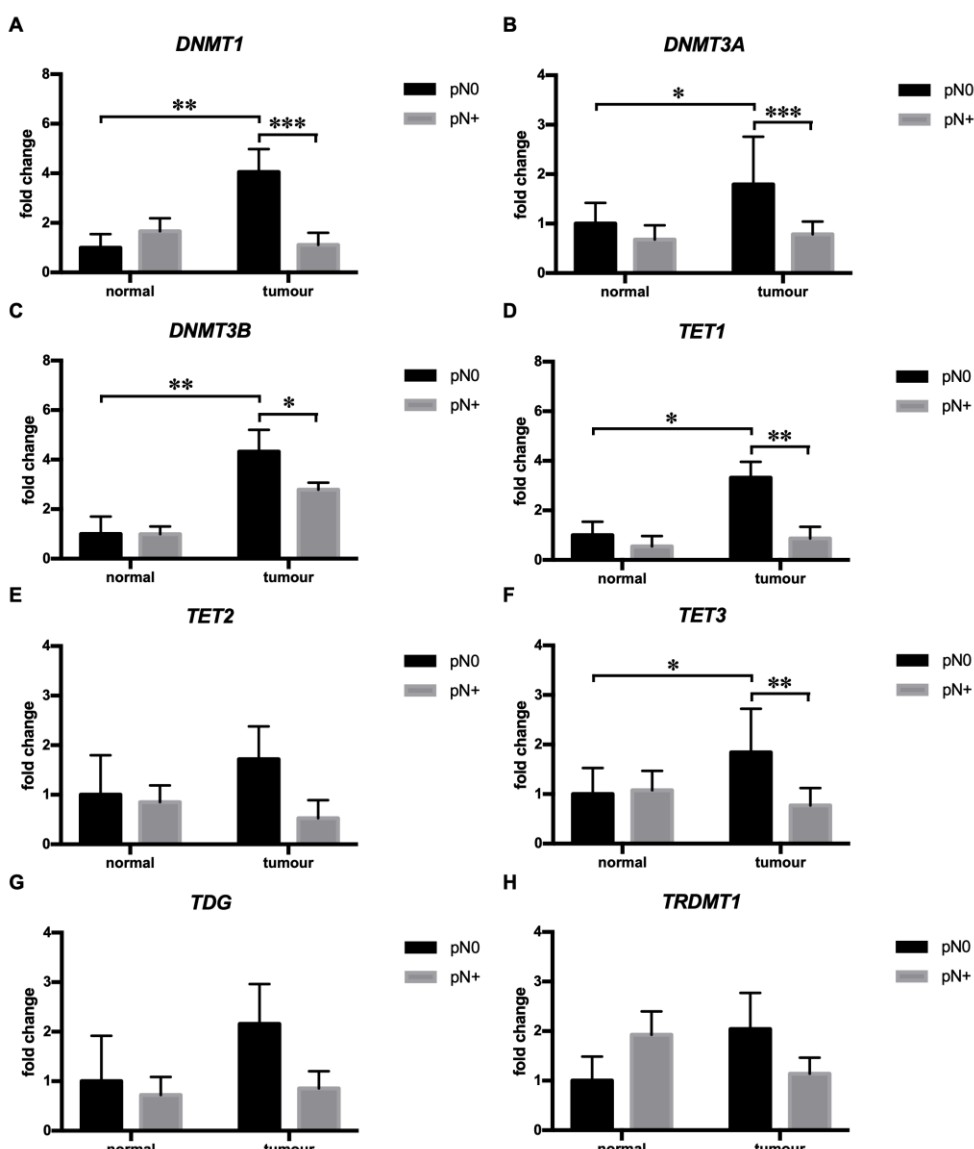

**Figure 1.** Tumors with regional lymph node metastases were associated with diminished expression of DNMT1, 3A and 3B, TET2, and TDG. For each gene, paired normal tissue corresponding to pN0 tumors was used as a reference. Normal/tumor pN0 (*n* = 7), normal/tumor pN+ (*n* = 24). The data are shown as the mean +/− S.E.M., * $p < 0.05$, ** $p < 0.01$, and *** $p < 0.001$. pN0—tumors without regional lymph node metastases; pN+—tumors with regional lymph node metastases.

*3.2. Perivascular Invasion (PVI) Was Associated with Diminished DNMT3B mRNA Expression in HNSCC*

Next, we analyzed the expression profile of methyltransferases/demethylases in relation to tumor invasion into the surrounding tissue. In tumors with perivascular invasion (PVI+), the expression of DNMT3B was significantly ($F_{(1, 30)}$ = 5.817, $p < 0.05$) higher in tumors than the adjacent normal tissue. Post hoc analysis revealed that DNMT3B expression was significantly ($p < 0.01$) higher in local tumors than in adjacent normal tissue. In contrast, the difference did not reach significance in PVI+ tumors due to decreased

DNMT3B expression in tumor tissue, indicating that PVI is driven in the absence of DNMT3B (Figure 2C). The expression of DNMT1, DNMT3A, TET1, TET2, TET3, TDG, and TRDMT1 was not associated with PVI in HNSCC (Figure 2). Similarly, the expression profile of methyltransferases/demethylases did not differ between tumors with lymphovascular and/or perineural invasion (Figures S1 and S2).

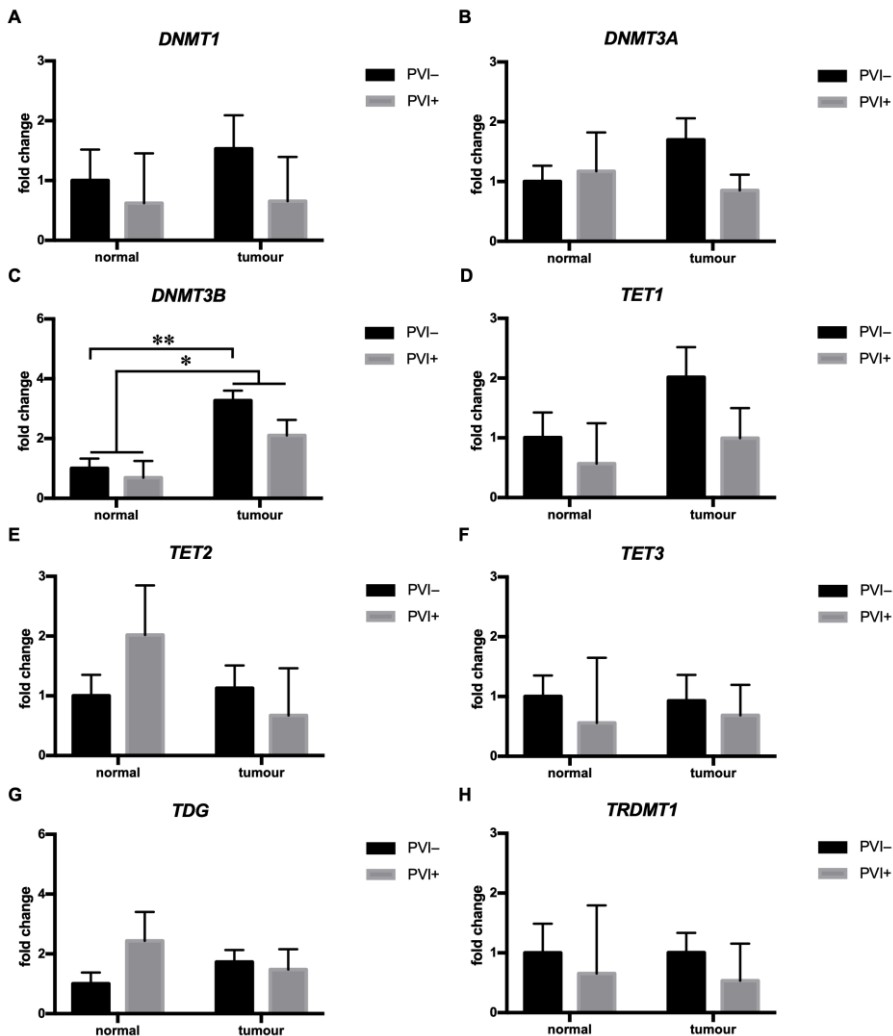

**Figure 2.** Tumors with perivascular invasion showed diminished DNMT3B expression. For each gene, paired normal tissue corresponding to local tumors was used as a reference. Normal/tumor PVI− ($n = 24$); normal/tumor PVI+ ($n = 8$). The data are shown as the mean +/− S.E.M. * $p < 0.05$; ** $p < 0.01$.

### 3.3. CpG73 Methylation Was Associated with the Diminished mRNA Expression of DNMT3B, TET2, and TDG in HNSCC

Our previous study showed increased CpG73 methylation in HNSCC [43]. To further our knowledge of the regulation of CpG73 methylation, the present study correlated the level of CpG73 methylation with the expression profile of DNMT1, DNMT3A, DNMT3B, TET1, TET2, TET3, TDG, and TRDM1 in tumor and normal tissues. In HNSCC tumor tissues, increased CpG73 methylation levels were in correlation with decreased mRNA expression of TET2 ($p < 0.05$) and TDG ($p < 0.01$) (Figure 3). Similarly, increased expression of DNMT3B was found in tumors with decreased CpG73 methylation levels, indicating that DNMT3B may not be involved in the methylation of CpG73 in HNSCC. TRDMT1 expression showed an inverse correlation with CpG73 methylation levels; however, the correlation did not reach significance ($p = 0.060$). The expression of DNMT1, DNMT3A,

TET1, and TET3 was not associated with CpG73 methylation in HNSCC tumors (Figure 3). In normal tissue, the expression of the methyltransferases and demethylases studied did not correlate with the level of CpG73 methylation (Figure S3).

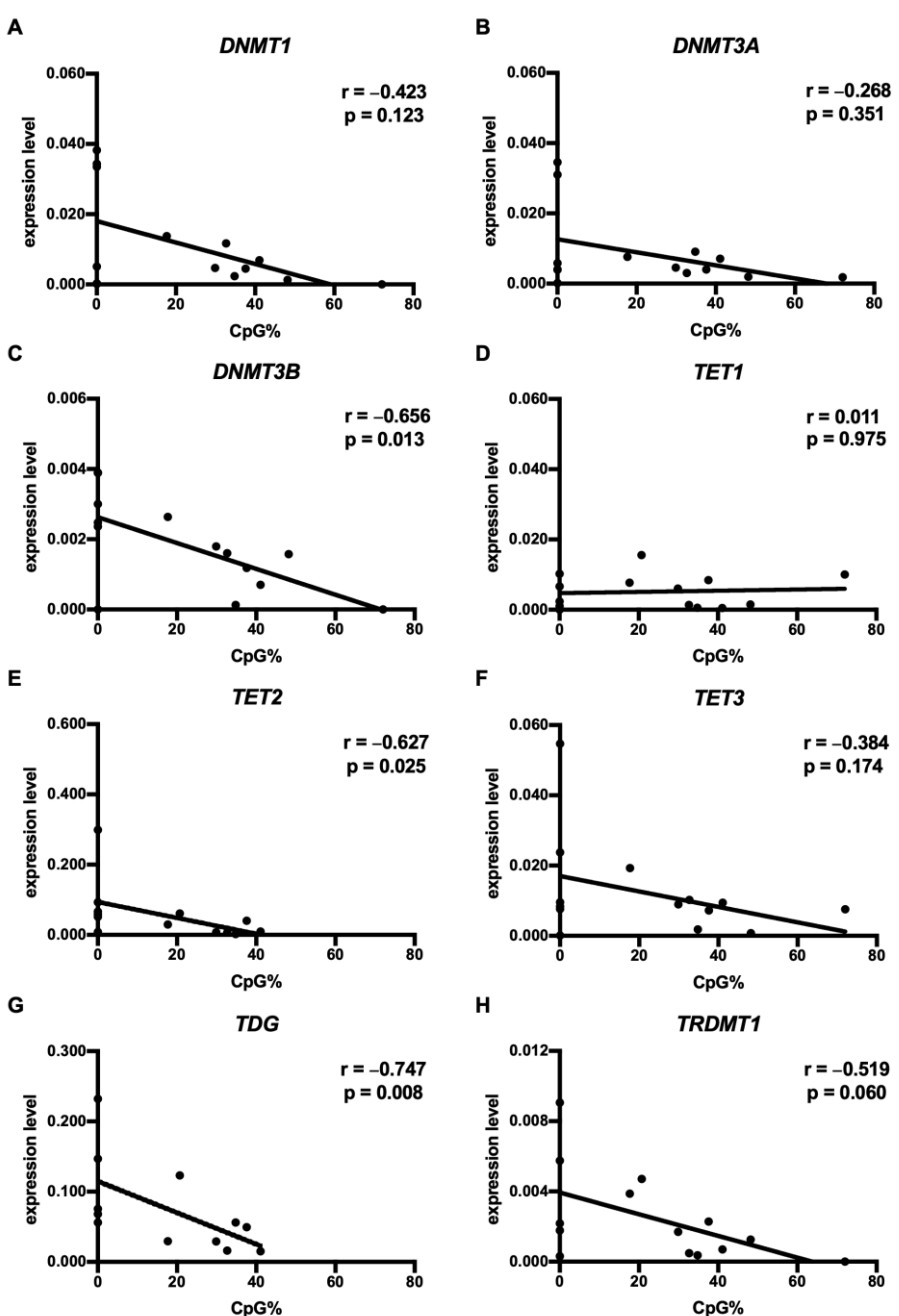

**Figure 3.** The level of CpG73 methylation was in negative correlation with the expression of DNMT3B, TET2, and TDG in HNSCC. r = Spearman's rank correlation coefficient for TDG: *n* = 15; for other genes: *n* = 16.

### 3.4. HPV16 Was Associated with Increased DNMT3B mRNA Expression in HNSCC

We determined the expression profile of the DNA methyltransferases DNMT1, DNMT3A, and DNMT3B, the DNA demethylases TET1, TET2, and TET3, and the RNA methyltransferase TRDMT1 in HNSCC tumors driven by HPV16. Among all of the genes studied, HPV16 was only associated with the expression of DNMT3B, which was significantly ($F_{(1, 18)}$ = 9.249, $p < 0.01$) higher in tumors compared to paired normal tissue, and this difference was a result of significantly ($p < 0.05$) higher expression of DNMT3B

in the tumor tissue compared to normal tissue in HPV16(+) but not HPV16(−) tumors, as revealed by Sidak's multiple comparison post hoc test (Figure 4C). Similarly, higher expression in the tumor tissue than the normal tissue was found for DNMT3A; however, the difference did not reach significance ($p = 0.09$) (Figure 4A). HPV16 did not affect the expression of DNMT1, TET1, TET2, TET3, TDG, and TRDMT1 in HNSCC (Figure 4).

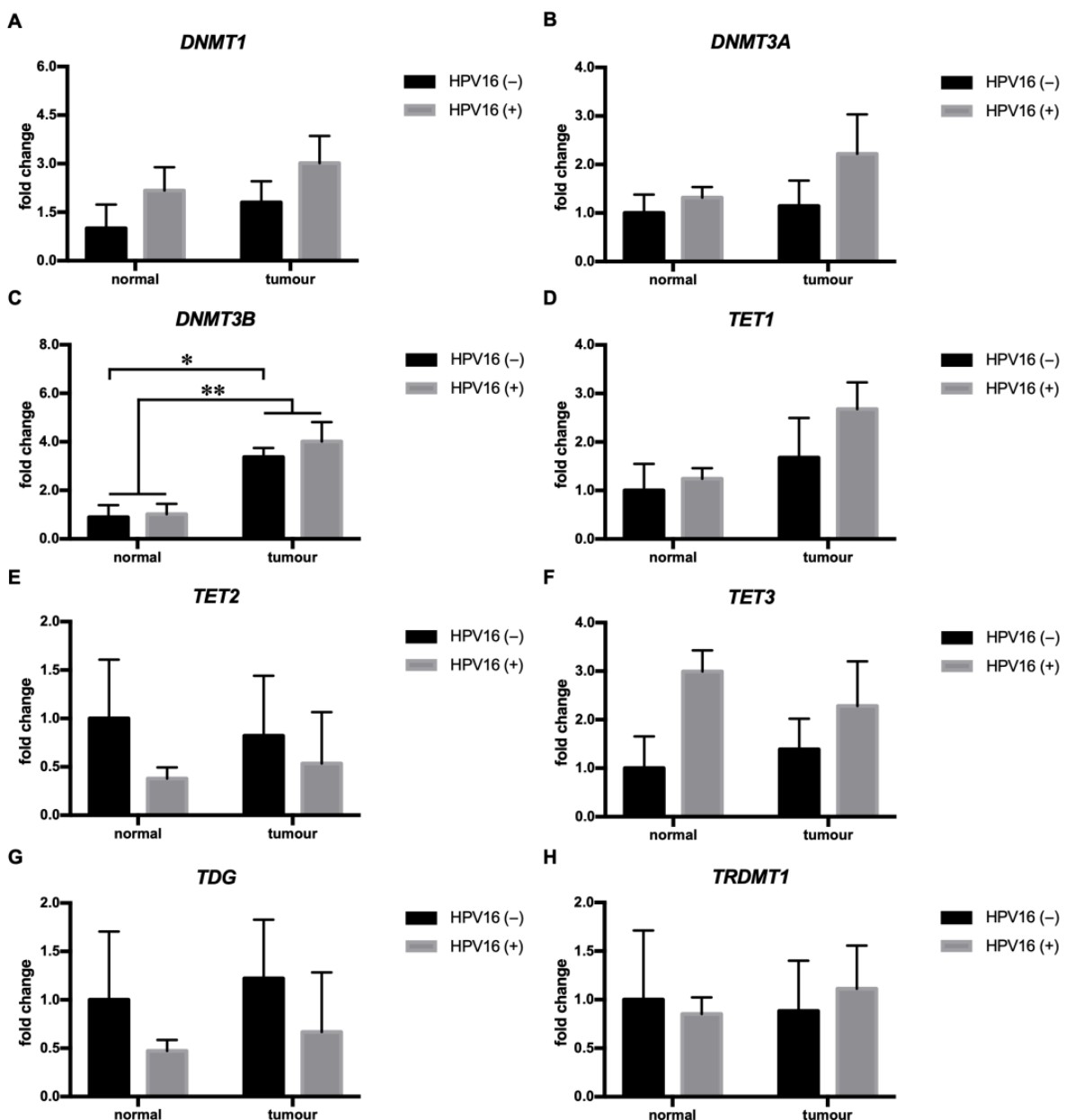

**Figure 4.** HPV16(+) tumors showed altered DNMT3B expression. For each gene, paired normal tissue corresponding to HPV16(−) tumors was used as a reference. Normal/tumor HPV16(−) ($n = 14$); normal/tumor HPV16(+) ($n = 6$). The data are shown as the mean +/− S.E.M. * $p < 0.05$; ** $p < 0.01$.

### 3.5. HNSCC Tumors Exhibited Higher DNMT1, 3A and TDG Protein Expression

To gain a better understanding of the role of the studied methyltransferases/demethylases in HNSCC as shown by their mRNA expression profiles, we determined the expression of the DNA methyltransferases DNMT1 and 3A and the DNA demethylase TDG at the protein level by in silico analysis. The protein expression of DNMT1, 3A, and TDG was significantly higher in tumors compared to normal tissue (DNMT1, 3A, and TDG: $p < 0.001$) (Figure 5A,C,E). Similarly, protein expression of the investigated enzymes was

significantly higher in stage 1 (DNMT1, 3A: $p < 0.05$; TDG: $p < 0.01$), 2 (DNMT1, TDG: $p < 0.001$; DNMT3A: $p < 0.01$), 3 (DNMT1, 3A, TDG: $p < 0.001$), and 4 (DNMT1, 3A, TDG: $p < 0.001$) tumors than in normal tissue (Figure 5B,D,E). When DNMT1, 3A, and TDG protein expression were compared across the tumor stages, DNMT3A protein expression was higher in stage 4 tumors compared to stage 2 tumors alone (Figure 5C)

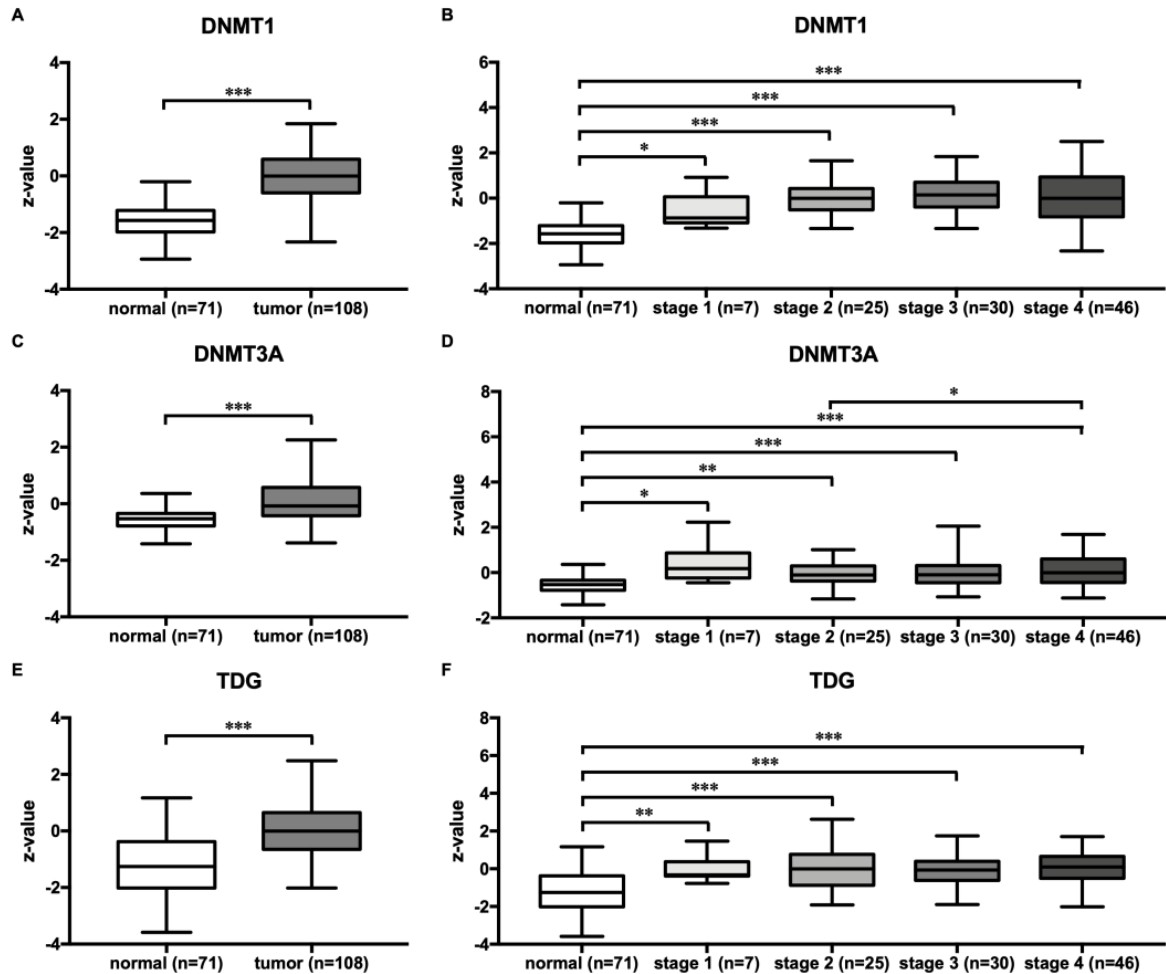

**Figure 5.** Protein expression of DNMT1, 3A, and TDG in HNSCC tumors. Gene expression is shown for tumors and normal tissue (**A,C,E**) and for tumors classified by tumor stage (1 to 4) and normal tissue (**B,D,F**). Protein analysis was performed by UALCAN using mass spectrometry data from the Clinical Proteomic Tumor Analysis Consortium (CPTAC) and the International Cancer Proteogenome Consortium (ICPC) datasets. The data are shown as box plots. * $p < 0.05$, ** $p < 0.01$, and *** $p < 0.001$.

## 4. Discussion

In the present study, we characterized the gene expression profile of the DNA methyltransferases *DNMT1*, *DNMT3A*, and *DNMT3B*, the DNA demethylases *TET1*, *TET2*, *TET3*, and *TDG*, and the RNA methyltransferase *TRDMT1* to gain a more comprehensive insight into the epigenetic regulation of HNSCC aggressiveness and HPV-driven tumorigenesis by DNA and RNA methylation.

HNSCC aggressiveness and reduced patient survival have previously been associated with the aberrant expression of DNMT1, DNMT3A, and TET genes [16,17,19,47]. To better understand the epigenetic control of cell invasion and metastasis in HNSCC, in the present study, we examined the expression profile of DNA methyltransferases/demethylases in tumors with perineural invasion (PNI), perivascular invasion (PVI), and/or lymphovascular invasion (LVI) and in tumors with regional lymph node metastases (pN+).

Epigenetic reprogramming of the genome, regulated by the interplay of various methyltransferases and demethylases occurs during tumor development [8,9,11]. Hypomethylation of CpG islands activates previously silenced proto-oncogenes, thus promoting carcinogenesis. On the other hand, hypermethylation of promotor regions leads to impaired transcription of tumor suppressor genes that regulate the cell cycle, metabolism, differentiation, cell death, angiogenesis, and metastasis [48]. Here, we show increased expression of the DNA-methyltransferases DNMT1, 3A, and 3B and the demethylases TET1 and 3 in tumors in comparison to paired normal tissue, while no differences were found for TET2 and TRDMT1. Furthermore, pN0 tumors exhibit increased expression of DNMT1, 3A and 3B, and TET1 and 3 compared to paired normal tissue. Interestingly, the expression of DNMT1, 3A and 3B, and TET1 and 3 did not differ between pN+ tumors and paired normal tissue and was significantly decreased compared to pN0 tumors. A distinct expression profile of DNA methyltransferases/demethylases in pN+ tumors suggests that following the initial reprogramming of DNA methylation patterns (expression profile of pN0), the development of regional lymph node metastases may require additional DNA methylation pattern alterations (expression profile of pN+) in solid tumors. Furthermore, our results suggest that these changes are directed towards global hypomethylation, which may be necessary to induce the expression of pro-metastatic genes through specific transcriptional events [49]. Finally, our study supports previous observations obtained by the in silico analysis of HNSCC samples from The Cancer Genome Atlas (TCGA) dataset indicating a significant correlation between the tumors with low DNMT1 expression and poor survival [50].

Protein analysis showed increased expression of DNMT1, DNMT3A, and TDG in tumors compared to normal tissue, which is consistent with our gene expression data and further supports our initial observations. Moreover, DNMT3A exhibited a stage-dependent expression pattern and showed an increase in protein expression in stage 4 compared with stage 2. DNMT1 and TDG protein levels, on the other hand, showed no differences across tumor stages. Interestingly, a recent IHC study also analyzed the protein expression of DNMT1 and DNMT3A in relation to tumor stage. This study revealed significantly increased expression of DNMT1 in lower tumor stages (I and II) compared with later stages (III and IV), whereas no significant difference in expression was found for DNMT3A between tumor stages [51]. The conflicting results may be attributed in part to different methods, such as the use of mass spectrometry (MS) or immunohistochemistry (IHC) to measure protein expression and highlight the need for further comprehensive studies to conclusively clarify the role and expression patterns of DNMT1, DNMT3A, and TDG at different stages of tumor development.

Recent evidence has shown that tumor invasion into neighboring tissue is regulated by various methyltransferases, such as DNMT1 and DNMT3B [23,24]. Our data are in line with previous observations and provide new evidence that the expression of DNMT3B is specifically altered in PVI+ tumors but not in PNI+ and LVI+ tumors. DNMT3B expression was higher in PVI− tumors than paired normal tissue, while the difference between PVI+ tumors and paired normal tissue did not reach significance due to the decreased DNMT3B expression in the tumors, indicating DNMT3B expression/activity was altered in PVI+ compared to PVI− HNSCC. Whether the observed differences in DNMT3B expression between PVI+ and PVI- HNSCC are reflected in the methylation patterns is unknown and warrants further investigation. Nevertheless, to the best of our knowledge, our study is the first to describe altered DNMT3B expression in relation to PVI in HNSCC. Of interest, DNMT3B overexpression has previously been associated with cell migration, invasion, and metastasis and epithelial–mesenchymal transition (EMT) by suppressing TIMP3 and E-cadherin, respectively, as suggested by in vitro studies using HNSCC cell lines [19,24]. Moreover, DNMT3B expression has been associated with vascular invasion in sporadic human renal cell carcinoma [52], further suggesting the role of DNMT3B in tumor invasion. Unlike perivascular invasion, expression of the studied methyltransferases and demethylases was not associated with perineural and lymphovascular invasion in HNSCC.

We have previously demonstrated that CpG73 hypermethylation in HNSCC is associated with decreased miR-2682 expression and shorter disease-free survival, suggesting a role for CpG73/miR-2682 in HNSCC aggressiveness [43]. Similarly, miR-2682 has been suggested to act as a tumor suppressor in osteosarcoma and pancreatic cancer [53,54]. Here, we show that CpG73 methylation levels are inversely correlated with the expression of the DNA demethylases TET2 and TDG, which may be responsible for CpG73 hypermethylation in tumor cells, in line with previous observations that the inactivation of DNA demethylases contributes to DNA hypermethylation often observed in cancer [55]. Interestingly, CpG73 methylation levels were also inversely correlated with DNMT3B expression, suggesting that CpG73 hypermethylation in tumor cells is determined independently of DNMT3B. Overall, our study suggests that DNMT1, 3A, and 3B are not biological determinants of CpG73 hypermethylation in tumor cells. Further investigation is warranted to determine the epigenetic mechanisms underlying the CpG73 methylation levels in HNSCC.

HPV-driven HNSCC tumorigenesis depends on the action of HPV E6 and E7 oncogenes that affect the expression of tumor-suppressor genes, leading to tumorigenesis [25–29]. Amongst the affected genes are additionally the DNA methyltransferases DNMT1, 3A, and 3B, whose mRNA expression and/or activity was increased by HPV oncogenes as shown in in vitro studies using cell lines harboring HPV [30–34]. In line with in vitro studies, a growing body of evidence has shown that HPV-positive and -negative HNCSS differ in methylation patterns on a global scale. For example, genome-wide hypomethylation, as measured by methylation levels of LINE and Alu elements, was more pronounced in HPV-negative than HPV-positive HNSCC [35]. In another study, unsupervised clustering analysis of 1505 CpG sites across 807 genes in 68 HNSCC tumor samples revealed differences in methylation patterns between HPV-positive and HPV-negative tumors [36]. In a recent study, a global analysis of 63 cases of HPV-positive and 263 HPV-negative HNSCC tumors identified 4371 hypermethylated and 2044 hypomethylated regions associated with HPV status. Moreover, the same study also revealed that 60% of differentially methylated genes that were hypermethylated in HPV-negative tumors were hypomethylated in HPV-positive tumors [38]. Altogether, these data indicate that tumorigenesis in HPV-positive HNSCC takes place in the presence of distinct DNA methylation reprogramming regulated by HPV oncogenes. In our study, we found a difference in DNMT3B expression between HPV16(+) and HPV16(−) tumors, with HPV16(+) tumors exhibiting higher DNMT3B expression than paired normal tissue, while the difference did not reach significance in HPV16(−) tumors. Our results are in line with a previous observation showing increased DNMT3B expression in OSCCC [56], and together with an in vitro study on primary human keratinocytes [34], they indicate the role of DNMT3B in HPV-associated malignancies, including HNSCC.

In summary, the studies have confirmed that global reprogramming of DNA methylation patterns plays a critical role in HNSCC development and progression. Our results, although based on a small sample size, illustrate that HNSCC tumors with perivascular invasion and regional lymph node metastases are characterized by a specific expression profile of DNA methyltransferases/demethylases compared to local tumors. This strengthens the rationale for further investigation of the interplay between DNA-modifying enzymes in HNSCC tumorigenesis. We acknowledge that our study's limited sample size is a key limitation and could affect the statistical power of our findings. Therefore, our observations need to be validated in independent cohorts to define novel potential prognostic biomarkers and molecular therapeutic targets confidently.

**Supplementary Materials:** The following supporting information can be downloaded at: https://www.mdpi.com/article/10.3390/cimb45060294/s1.

**Author Contributions:** Conceptualization, L.G., T.B. and U.P.; methodology, L.G. and H.Č.; software, T.B.; validation, L.G. and T.B.; data curation, H.Č.; resources, M.Š. and B.L.; writing—original draft preparation, L.G. and T.B.; writing—review and editing, L.G., T.B., U.P., M.Š., H.Č. and B.L.;

visualization, L.G. and T.B.; supervision, U.P.; project administration, U.P.; funding acquisition, U.P. and B.L. All authors have read and agreed to the published version of the manuscript.

**Funding:** This research was funded by the Slovenian Research Agency research core funding P3-0427 and P3-0067. This research was also supported by University Medical Center Maribor internal grant no. IRP-2021/02-14 and no. IRP-2015/01-21.

**Institutional Review Board Statement:** The study was approved by the Medical Ethics Committee of UMC Maribor (reference number UKCMB-KME11-5/15) and conducted according to the Declaration of Helsinki.

**Informed Consent Statement:** Informed consent was obtained from all subjects involved in the study.

**Data Availability Statement:** Not applicable.

**Acknowledgments:** The authors thank the patients for donating their clinical samples and the nursing personnel at the University Medical Centre Maribor for their assistance in collecting the clinical samples.

**Conflicts of Interest:** The authors declare no conflict of interest.

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
