# Peer review of "Gene Expression Profiles of Methyltransferases and Demethylases Associated with Metastasis, Tumor Invasion, CpG73 Methylation, and HPV Status in Head and Neck Squamous Cell Carcinoma"

_cimb, doi:10.3390/cimb45060294_

Round 1

Reviewer 1 Report

Comments and Suggestions for Authors

Uroš Potočnik et al. examined the mRNA expression of various molecules, such as DNMT1, DNMT3A/B, TET1/2/3, TDG and TRDMT1. The author found that the relationship between these molecules and the clinical-pathological parameters. In the all, the innovation of this paper is limited.

1.      The samples size of this paper is limited, especially for women.

2.      The author should confirm the result using other methods, such as western blot or immunohistochemistry.

3.      The author may confirm the result using the bioinformatic analysis.

4.      In the figure 1, why the number of pN+ is 24, which is not consistent with Table1.

5.      The positive result of the paper is not enough.

Reviewer 2 Report

Comments and Suggestions for Authors

In this manuscript authors used a cohort on HNSCC patients to investigate the expression of many DNA methyltransferases and demethylases and their association with tumour stage, metastasis and DNA methylation. For what concern DNA methylation, authors show, in particular, results on CpG73 methylation, but it is not described how they have obtained those data. Moreover, figures 1 and 4 show expression of different genes as fold changes, taking as a reference the paired normal tissue corresponding to pN0 for figure 1 or to HPV- for figure 4. I think that each tumour has to be related to the normal counterpart (tumour pN+ to normal pN+ counterpart and tumour HPV+ to normal HPV+ counterpart). Anyway, since authors show the results as fold change over normal pN0 or HPV-, these values should be all 1-fold change, instead, as shown in figure 1G, 1E or 4E, 4G and 4H, this is not the case. Thus, authors should explain how these measures were calculated, similarly also in figure 2. 

Reviewer 3 Report

Comments and Suggestions for Authors

Through this study, the authors tried to find a correlation between the expression profile of some methyltransferases and demethylases and the level of CpG73 methylation. In addition, they attempted to identify the effect of perivascular invasion and HPV16 on DNMT3B methyltransferase expression in HNSCC. The data support the importance of studied DNA methyltransferases and demethylases as potential prognostic biomarkers and molecular therapeutic targets for HNSCC.

This study is ingeniously designed, and the results are presented and compared to similar studies in the literature. The conclusions are supported by the data presented, this paper being of interest considering the original approach on the scorching topic addressed.

There are some typographical errors in the manuscript that I have highlighted. Some sentences that I have highlighted should be rewritten for clarity.

Also, Figure 1 should be mentioned in the results or discussions chapter. Also, I suggest improving the resolution of the presented figures or choosing another way of presentation to make it easier to follow.

Overall, the article could be a valuable contribution to the journal. So, I would recommend the manuscript for publishing after minor changes and updates have been made by the authors.

Reviewer 4 Report

Comments and Suggestions for Authors

Abstract: was well written

Introduction: was comprehensive and informative.

The materials and Methods were well designed, please note that it was reported the methylation rate in the HPV16 L1 gene was higher for high-grade CIN (≥CIN2/high-grade squamous intra-epithelial lesion (HSIL) (95% confidence interval (95%CI:72·7% (47·8–92·2))) vs. low-grade CIN (≤CIN1/low-grade squamous intra-epithelial lesion (LSIL) (44·4% (95%CI:16·0–74·1). They concluded Higher HPV methylation is associated with increased disease severity ( refer. A), If the methylation rate of the HPV 16 L1 gene was compared with different tumor staging, and also in tumor invasiveness and metastasis, it should have been given better outcomes.

The results were well described.

The discussion was informative and constructive.

Novelty, they have identified the effect of perivascular invasion and HPV16 on DNMT3B expression in HNSCC, and the expression of TET2 and TDG was inversely correlated with the hypermethylation of CpG73.

They found HPV16 was only associated with the expression of DNMT3B, which was significantly  higher in tumors compared to paired normal tissue. HPV16 did not affect the expression of DNMT1, TET1, TET2, TET3,TDG, and TRDMT1 in HNSCC.

Round 2

Reviewer 2 Report

Comments and Suggestions for Authors

Authors answered to all the raised questions.